# Primary Ciliogenesis by 2-Isopropylmalic Acid Prevents PM2.5-Induced Inflammatory Response and MMP-1 Activation in Human Dermal Fibroblasts and a 3-D-Skin Model

**DOI:** 10.3390/ijms222010941

**Published:** 2021-10-10

**Authors:** Ji-Eun Bae, Daejin Min, Ji Yeon Choi, Hyunjung Choi, Joon Bum Kim, Na Yeon Park, Doo Sin Jo, Yong Hwan Kim, Hye-Won Na, Yoon Jae Kim, Eun Sung Kim, Hyoung-June Kim, Dong-Hyung Cho

**Affiliations:** 1Brain Science and Engineering Institute, Kyungpook National University, Daegu 41566, Korea; loveg730@naver.com; 2R&D Center, AMOREPACIFIC Corporation, Yongin 17074, Korea; djmin@amorepacific.com (D.M.); heart@amorepacific.com (H.C.); serina@amorepacific.com (H.-W.N.); 3School of Life Sciences, BK21 FOUR KNU Creative BioResearch Group, Kyungpook National University, Daegu 41566, Korea; muse41191@naver.com (J.Y.C.); kss3213@naver.com (J.B.K.); yeonie5613@gmail.com (N.Y.P.); doosinjo@gmail.com (D.S.J.); yoo035913@gmail.com (Y.H.K.); chodong02@hanmail.net (Y.J.K.); 4New Drug Development Center, Osong Medical Innovation Foundation, Cheongju 28160, Korea; eskim@kbiohelth.kr

**Keywords:** 2-IPMA, primary cilia, dermal fibroblasts, oxidative stress, inflammation

## Abstract

Particulate matters (PMs) increase oxidative stress and inflammatory response in different tissues. PMs disrupt the formation of primary cilia in various skin cells, including keratinocytes and melanocytes. In this study, we found that 2-isopropylmalic acid (2-IPMA) promoted primary ciliogenesis and restored the PM2.5-induced dysgenesis of primary cilia in dermal fibroblasts. Moreover, 2-IPMA inhibited the generation of excessive reactive oxygen species and the activation of stress kinase in PM2.5-treated dermal fibroblasts. Further, 2-IPMA inhibited the production of pro-inflammatory cytokines, including IL-6 and TNF-α, which were upregulated by PM2.5. However, the inhibition of primary ciliogenesis by IFT88 depletion reversed the downregulated cytokines by 2-IPMA. Moreover, we found that PM2.5 treatment increased the MMP-1 expression in dermal fibroblasts and a human 3-D-skin model. The reduced MMP-1 expression by 2-IPMA was further reversed by IFT88 depletion in PM2.5-treated dermal fibroblasts. These findings suggest that 2-IPMA ameliorates PM2.5-induced inflammation by promoting primary ciliogenesis in dermal fibroblasts.

## 1. Introduction

Fine particulate matters (PMs, (PM2.5 = diameters < 2.5 μm)) in air pollution are heterogeneous pollutants composed of several molecules. Although numerous studies have shown that PM2.5 is associated with cardiovascular diseases and pulmonary diseases by causing endothelial dysfunction and epithelium injury [1,2], the patho-physiological effect of PM2.5 on the skin has been limitedly elucidated. However, the interest in PMs on the skin has greatly increased and recent epidemiological studies indicate PM2.5 disrupts skin homeostasis [3]. PMs can penetrate the epidermis in normal intact skin and cause cutaneous diseases, such as atopic dermatitis and psoriasis, by increasing oxidative stress and inflammatory response [4].

Primary cilia function as antennae for sensing extracellular circumstances and mediate the interactions of cells and external stimuli [5,6]. Cilia as microtubule-based cellular organelles are maintained by an intraflagellar transport proteins (IFTs) and dysregulation of ciliogenesis is associated with various human diseases, known as ciliopathies [5,6]. As skin is the outermost organ covering the body, it has to sense various extracellular environment. Thus, primary cilia in skin cells can play various roles including modulation of oxidative stress and inflammatory response [7]. Consistently, primary cilia have been identified in skin cells, such as keratinocytes, melanocytes, fibroblasts, basal cell carcinoma, and Langerhans cells [7]. Dermal primary cilia are involved in hair follicle development by regulating sonic hedgehog signaling [8]. Our group recently reported that primary cilia inhibit skin pigmentation in melanocytes by activating smo-Gli2 signaling [9]. Moreover, we showed that PM2.5 decreases primary ciliogenesis and cell differentiation by regulating the JNK pathway in human epidermal keratinocytes [10].

Although primary cilia are important for development and maintenance in the dermis, the role of primary cilia in response to dermal toxic PM2.5 has not been elucidated yet. We identified 2-isopropylmalic acid (2-IPMA) as a potent inducer for primary ciliogenesis from a high content screening with a metabolite library. 2-IPMA is an intermediate metabolite in leucine biosynthesis, which is found in metabolite profiles of white wines [11,12]. Previously, it was reported that 2-IPMA detoxified aluminum ions by direct chelation in yeast and 2-IPMA is an internal standard in the diagnostic model for cancer using human serum metabolomics [13,14]. However, the physiological role of 2-IPMA in skin cells has not been evaluated. Thus, the aim of this study was to evaluate 2-IPMA as an inducer for ciliogenesis and investigate 2-IPMA on PM2.5-mediated dysgenesis of primary cilia in dermal fibroblasts.

## 2. Result

## 2.1. 2-IPMA Restores PM2.5-Mediated Ciliary Dysgenesis in Normal Dermal Fibroblasts

From a cell-based high content screening with a microbiome metabolite library, 2-IPMA was identified as a strong inducer of primary ciliogenesis in retinal pigment epithelial cells. 2-IPMA is an intermediate metabolite in leucine biosynthesis and has antioxidant and antibacterial activities [15,16]. However, the role of 2-IPMA in the skin has not been investigated; thus, the effects of 2-IPMA as a ciliary inducer on human normal dermal fibroblasts were further determined. First, the cytotoxic activity of 2-IPMA was examined, and no cytotoxicity was found (~1000 µM) in dermal fibroblasts (Figure 1A). To observe primary cilia, we also employed ARL13B staining, which is a widely used maker for primary cilia [17]. To confirm the screening results, the ciliary response in dermal fibroblasts to 2-IPMA was determined. Consistent with the screening, 2-IPMA highly induced primary ciliogenesis in dermal fibroblasts (Figure 1B,C). A smoothened agonist (SAG) was used a positive control for ciliary inducer (Figure 1B,C). In addition, similar to dermal keratinocytes, we found that PM2.5 decreased both the ciliated cells and cilium lengths in dermal fibroblasts (Figure 2A,B). However, 2-IPMA notably restored the PM2.5-mediated decrease in primary ciliogenesis, suggesting that 2-IPMA inhibits PM2.5-mediated loss of primary cilia in dermal fibroblasts. Moreover, the depletion of the ciliary core protein IFP88 suppressed the 2-IPMA-induced primary ciliogenesis (Figure 2C–E), supporting that 2-IPMA induces primary ciliogenesis and restores PM2.5-mediated ciliary dysgenesis in dermal fibroblasts.

### 2.2. 2-IPMA Inhibits PM2.5-Induced JNK Activation and Inflammatory Response in Normal Dermal Fibroblasts

We recently reported that PM2.5 activates the JNK pathway, which mediates cellular oxidative stress and pro-inflammatory responses in dermal keratinocytes [10]. Therefore, the effects of 2-IPMA on JNK activation in PM2.5-exposed cells were further evaluated. Similar to previous results in keratinocytes, the treatment with PM2.5 in dermal fibroblasts induces JNK phosphorylation, which was blocked by the treatment with a JNK inhibitor, SP600125 (Figure 3A,B). Interestingly, treatment with 2-IPMA additionally inhibited JNK phosphorylation in PM2.5-treated cells (Figure 3A,B). The PM2.5-induced decrease in ciliogenesis was significantly rescued by 2-IPMA and SP600125 (Figure 3C,D), suggesting that the JNK pathway is also involved in ciliary dysgenesis caused by PM2.5 in dermal fibroblasts.

Since inflammatory responses contribute to PM2.5-mediated skin injury, we investigated the effects of 2-IPMA-induced primary ciliogenesis on inflammatory cytokine production. The levels of secreted IL-6 and TNF-α were measured in normal dermal fibroblasts treated with PM2.5 with or without 2-IPMA. The treatment with PM2.5 highly increased the expressions of both IL-6 and TNF-α, which were suppressed by either 2-IPMA or SP600125 in dermal fibroblasts (Figure 4A,B). In addition, inhibition of primary cilia by depletion of IFT88 reversed the downregulated cytokines by 2-IPMA or SP600125 in PM2.5-treated dermal fibroblasts (Figure 4A,B). Nuclear factor-κB (NF-κB), a transcriptional factor, is a central mediator of inflammation induced by exposure of PM2.5. Thus, we further examined the effect of 2-IPMA on activation of NF-κB in PM2.5-treated dermal fibroblasts. Consistently, 2-IPMA inhibited the activation of NF-κB by PM2.5 in dermal fibroblasts (Figure 4C). In addition, depletion of IFT88 also suppressed the inhibitory effect to 2-IPMA on NF-κB activation in PM2.5-treatd cells (Figure 4C). These results suggest that 2-IPMA has anti-inflammatory effects by promoting ciliogenesis in PM2.5-treated dermal fibroblasts.

### 2.3. 2-IPMA Downregulates MMP-1 Expression by PM2.5 in Normal Dermal Fibroblasts and a 3-D-Skin Model

It was recently reported that PM2.5 exposure enhances the expression of MMPs, which contribute to skin cell aging, including keratinocytes and fibroblasts [18,19]. Therefore, the effects of 2-IPMA on PM2.5-induced MMP-1 expression was determined. The treatment with PM2.5 in skin fibroblasts strongly increased the MMP-1 expression in dermal fibroblasts (Figure 5A,B). However, the increased MMP-1 expression by PM2.5 was dramatically suppressed by either 2-IPMA or SP600125 in dermal fibroblasts (Figure 5A). Then, the role of primary cilia on the expression of MMP-1 was further evaluated. Notably, the reduced MMP-1 expression by 2-IPMA was reversed by IFT88 depletion in PM2.5-treated dermal fibroblasts (Figure 5A,B). The skin is a complex tissue composed of various appendages and layers [20,21]. Therefore, we determined the effect of 2-IPMA on the skin using a full-thickness human three-dimensional (3-D)-skin model more similar to natural skin. First, full-thickness 3-D-skin models were reconstructed and treated with PM2.5 (100 µg/mL) in the absence or presence of 2-IPMA (50 or 100 µM). Then, the tissues were subjected to a histological examination. The hematoxylin and eosin (H&E) staining results showed that our 3-D-skin models had a normal skin structure composed of a well-stratified epidermis and fibroblasts containing dermis. Moreover, no feature of cytotoxicity was observed in all the groups (Figure 5C). Whereas, in the MMP-1 immunocytochemistry results, an increase in MMP-1 expression was observed according to the PM2.5 treatment and suppression efficacy of 2-IPMA (Figure 5C). These results were confirmed through the quantification of MMP-1 expression in the dermis using image analysis (Figure 5C).

## 3. Discussion

From a cell-based high content screening with a metabolite library, we identified 2-IPMA as a strong inducer for primary ciliogenesis [22]. 2-IPMA is an intermediate metabolite in leucine biosynthesis, synthesized from α-ketoisovalerate and acetyl-coenzyme A, a reaction catalyzed by 2-isopropylmalate synthase in mitochondria and exported into the cytosol [23]. It was reported that 2-IPMA has antioxidant and antibacterial activities. Yeast cells naturally release 2-IPMA into their environment to chelate aluminum ions and prevent entry to the cells [13,23]. In this study, we found that 2-IPMA promotes primary ciliogenesis, which inhibits the inflammatory response and MMP-1 expression by PM2.5 in dermal fibroblasts and a 3-D-skin model.

Various pollutions have a negative impact on skin function and heath. Several studies have shown that skin cells treated with different pollutants are characterized by increased production of pro-inflammatory factors, such as IL-1α, IL-6, IL-8, and PGE2 [24]. Therefore, chronic PM2.5 exposure is associated with extrinsic skin aging, such as wrinkles and hyperpigmentation [25]. Furthermore, PM2.5 causes various skin disorders, including atopic dermatitis and psoriasis, by increasing oxidative stress and inflammatory response [25,26]. For example, PM2.5 activates various stress kinase signaling pathways, which induce the expression of transcription factors, such as JNK, extracellular signal-regulated kinase (ERK), and p38-dependent NF-κB [27]. Consistent with this notion, Jia et al. showed that PM2.5 induces an inflammatory response by activating the STAT3/JNK pathway in lung epithelial cells [28]. Previously, we reported that PM2.5 inhibits primary ciliogenesis by increasing oxidative stress and JNK activation in normal human epidermal keratinocytes [10]. PM2.5 enhanced the expression of SPRR3 by transcriptional regulation, which suppresses the formation of primary cilia in keratinocytes whereas inhibition of JNK recovered primary cilia in PM2.5-treated keratinocytes [10]. Wang et al. also showed that PM2.5 triggers inflammation by increasing the ROS-MAPK and NF-κB signaling pathway [29]. Accordingly, we found that PM2.5 increases the production of inflammatory cytokines, IL-6 and TNF-α, and NF-κB activation in dermal fibroblasts (Figure 4). Inhibition of JNK by SP600125 recovered the inflammatory responses as well as JNK activation in PM2.5-treated dermal fibroblasts (Figure 3 and Figure 4). Similar with SP600125, 2-IPMA significantly ameliorated the increase of the pro-inflammatory cytokines and NF-κB activation by PM2.5 in dermal fibroblasts, which was reversed by loss of ciliogenesis by depletion of IFT88 (Figure 4). Furthermore, our findings suggest that 2-IPMA recovers the PM2.5-induced inflammatory response by promoting primary ciliogenesis in skin cells.

Recently, we showed that 2-IPMA reduces oxidative stress by inhibition of excessive ROS generation by PM2.5 [22]. In addition, it was reported that 3-IPMA has free radical scavenging activity [16]. Various metabolites and natural compounds, such as monomers, phenolic compounds, and flavonoids, reduce cellular oxidative stress and the inflammatory response of PM2.5 in various cell types, including keratinocytes and fibroblasts [30,31,32]. The activation of the aryl hydrocarbon receptor (AhR) by environmental pollutants increases MMP-1 and MMP-3 expression in fibroblasts and hyperpigmentation in keratinocytes [33,34]. Resveratrol, a well-known polyphenol metabolite, inhibits the PM-induced inflammatory response in human keratinocytes by inhibiting the AhR/ROS/MAPK pathway, which promotes epithelial ciliogenesis [27,35]. Since 2-IPMA also inhibits PM-induced inflammation by promoting primary ciliogenesis, further study on the AhR and ROS pathway will help us understand the molecular mechanism of 2-IPMA.

Conclusively, during the last decade, the market for anti-pollution skincare and cosmetic products has grown significantly due to the worsening air quality. PM induces skin senescence, inflammatory skin diseases, and carcinogenesis by increasing oxidative and inflammatory stresses [36]. Although the molecular mechanism should be further explored, we found that 2-IPMA is a potent inducer of primary ciliogenesis and efficiently suppress PM2.5-induced cellular stress in dermal fibroblasts. Therefore, our results suggest that 2-IPMA is a potential cosmetic functional ingredient for anti-pollution skincare.

## 4. Material and Method

### 4.1. Cell Culture

Human dermal fibroblasts (GM08333) were purchased from the Coriell Institute for Medical Research (Camden, NJ, USA). The fibroblasts were cultured in Dulbecco’s Modified Eagle’s medium supplemented with 10% fetal bovine serum and 1% penicillin-streptomycin (Invitrogen, Carlsbad, CA, USA). For the 3-D reconstruction of human skin tissue, human neonatal dermal fibroblast (HDFn) was cultured in Medium 106 (M106) supplemented with low-serum growth supplement (LSGS). Human neonatal epidermal keratinocyte (HEKn) was cultured in EpiLife medium supplemented with human keratinocyte growth supplement (HKGS). Penicillin (100 U/mL) and streptomycin (100 mg/mL) were added to the medium as antibiotics, and the cells were cultured at 37 °C in an atmosphere containing 5% CO_2_. All cells, media, supplements, and antibiotics were purchased from Thermo Fisher Scientific (Waltham, MA, USA).

### 4.2. Reagents and Cell Viability

PM2.5 was collected on a Teflon filter using a low-volume air sampler consisting of a cyclone (2.5-μm size cut, URG-2000-30EH), two upstream denuders (annular, URG-2000-30 × 242-3CSS), Teflon filter (Zefluor^TM^ 2.0 μm Pall Life Sciences, Mexico city, Mexico), backup filter, and backup denuder in the series. The filter was sonicated in ethanol (EtOH) for 30 min, the EtOH was evaporated, and then the PM2.5 was resuspended in deionized water. The PMs were collected using a high-capacity air collector with a quartz filter. Chemical reagents, including Hoechst 33342, SP600125, 2-IPMA, and SAG, were purchased from Sigma-Aldrich (St. Louis, MO, USA). Cell viability was assayed using a Cell Counting Kit-8 (Dojindo Laboratories, Kumamoto, Japan) following the manufacturer’s protocol.

### 4.3. Western Blot Analysis and Densitometry Analysis

All lysates were prepared in 2 × Laemmli sample buffer (62.5 mM Tris-HCl, pH 6.8, 25% [*v*/*v*] glycerol, 2% [*w*/*v*] sodium dodecyl sulphate [SDS], 5% [*v*/*v*] β-mercaptoethanol, and 0.01% [*w*/*v*] bromophenol blue, Bio-Rad, Hercules, CA, USA). All cellular proteins were quantified using the Bradford solution (Bio-Rad) according to the manufacturer’s instructions. The samples were then separated using SDS-polyacrylamide gel electrophoresis (PAGE) and transferred onto a polyvinylidene fluoride (PVDF) membrane (Bio-Rad). After blocking with 4% (*w*/*v*) skim milk in Tris-buffered saline plus Tween (TBST; 25 mM Tris, 140 mM sodium chloride [NaCl], and 0.05% [*v*/*v*] Tween^®^ 20), the membranes were incubated overnight with the following specific primary antibodies: actin (1:10,000, EMD Millipore, MA, USA), IFT88 (1:3000, Proteintech, Chicago, IL, USA), MMP-1 (1:3000, Abcam, Cambridge, MA, USA), phospho-JNK (1:1000, Cell Signaling, Danver, MA, USA), JNK (1:3000, Cell Signaling), p65 (1:1000, Cell Signaling), and phospho-p65 (1:1000, Cell Signaling). For protein detection, the membranes were incubated with horseradish peroxidase (HRP)-conjugated secondary antibodies and the signals were detected using a SuperSignal^®^ West Dura HRP detection kit (Pierce, Rockford, IL, USA).

Densitometry was performed on scanned immunoblot images using the AE9300 Ez-Capture MG Hours Image Saver HR image capture tool (ATTO, Tokyo, Japan). Relative band intensities were analyzed using the CS Analyzer 3 software (ATTO). The relative intensity value for each experimental band was calculated by normalizing the experimental intensity to the corresponding loading control.

### 4.4. Primary Cilia Staining and Counting

For the staining of primary cilia, cells were washed with cold phosphate-buffered saline (PBS) and fixed with 4% (*w*/*v*) paraformaldehyde (PFA), dissolved in PBS containing 0.1% (*v*/*v*) Triton X-100. Subsequently, the cells were blocked with PBS containing 1% bovine serum albumin (BSA), and incubated overnight at 4 °C with primary antibodies against ARL13B (1:1000, 17711-1-AP, Proteintech, Chicago, IL, USA) in 1% BSA. After washing, the cells were incubated with Alexa Fluor 488-conjugated secondary antibodies at room temperature (RT) for 1 h. Then, cells were treated with Hoechst 33342 dye (1:10,000, H3570, Thermo-Fisher, Waltham, MA, USA) for nuclear staining. Cilia images were observed using a fluorescence microscope. Cilia were counted in about 200 cells under each experimental condition (*n* = 3). The ciliated cell percentage was calculated as (total number of cilia/total number of nucleus at each image) × 100. Cilia lengths were measured using the Free-hand Line Selection Tool of Cell Sense Standards software (version 3.1) (Olympus Europa Holding GmbH, Hamburg, Germany) and the average cilium lengths were calculated. Analysis of graph data was performed with GraphPad Prism (version 8.3.0) (GraphPad Software, San Diego, CA, USA).

### 4.5. ELISA Assay

The cytokine levels of cell culture supernatants were determined using ELISA. Human IL-6 and TNF-α in cell culture supernatants were separately determined by ELISA kits from BD biosciences (San Jose, CA, USA) and Abcam (Cambridge, MA, USA) according to the manufacturer’s instructions.

### 4.6. Reconstruction of Full-Thickness Human 3-D-Skin Model

A full-thickness human 3-D-skin model was reconstructed through slight modification of our previously described protocols [37,38]. Briefly, dermal mixture was prepared by mixing Type-I collagen (3 mg/mL, Nitta gelatin, Osaka, Japan), reconstruction buffer (Nitta gelatin), DE concentrated culture solution (Nitta gelatin), fibrinogen (10 mg/mL, Sigma-Aldrich. St. Louis, MO, USA), aprotinin (0.4 TIU/mL, Sigma-Aldrich), and fibroblasts. Then, thrombin (0.625 U/mL, Sigma-Aldrich) was added to initiate polymerization of fibrinogen and the mixture was loaded to each insert of 12 mm Snapwell cell culture inserts (Corning Costar, NY, USA) and incubated for 2 h at 37 °C to allow polymerization. The dermal layer was then cultured in 106 media for 7 days at 37 °C and 5% CO_2_. Then, HEKns were seeded on dermis and cultured in EpiLife media. The next day, the media was changed with the CnT-3D-PR medium (CELLnTEC, Bern, Switzerland) and incubated for 1 day. The formation of epidermal layers was induced by the air–liquid interface (ALI) culture for 10 days.

### 4.7. Histological Examination

The tissues were fixed in 10% neutral-buffered formalin (BBC Biochemical, Mount Vernon, WA, USA) and embedded within paraffin. The paraffin-embedded samples were sliced into 5 μm sections and histological observation was performed following hematoxylin and eosin (H&E) staining. For immunohistochemistry, tissue sections were stained with an MMP-1 antibody (ab52631, Abcam, Cambridge, MA, USA) at 4 °C overnight. The expression level of MMP-1 in the dermis was quantified by using Image-Pro Plus 7 software (Media Cybernetics, Inc. Rockville, MD, USA).

### 4.8. Statistical Analysis

Data were obtained from three independent experiments (*n* = 3) and are presented as means ± standard error (SEM). The results were statistically evaluated using a one-way analysis of variance (ANOVA) using the statistical package for the social sciences (SPSS, software, version 21). The *p* value of the data was compared with the control and calculated by Student’s *t*-test (*p* values of * *p* < 0.05, ** *p* < 0.01, *** *p* < 0.005).

## Figures and Tables

**Figure 1 ijms-22-10941-f001:**
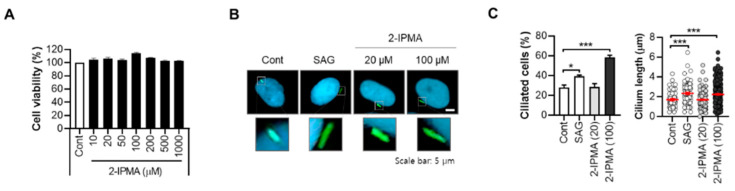
2-IPMA promotes primary ciliogenesis in normal dermal fibroblasts. (**A**) Normal dermal fibroblasts were treated with different concentration of 2-IPMA (10–1000 µM) and cell viability was determined by CCK-8 assay. (**B**,**C**) Normal skin fibroblasts were treated with either SAG (1 µM) or 2-IPMA (20, 100 µM). (**B**) Then, the cells were imaged by immuno-staining with ARL13B antibody (green) and Hoechst dye (blue). Scale bar: 5 µm. (**C**) The ciliated cells and cilium length of the cells were measured under a fluorescence microscope. Data are presented as the mean ± SEM (*n* = 100, * *p* < 0.05, *** *p* < 0.005).

**Figure 2 ijms-22-10941-f002:**
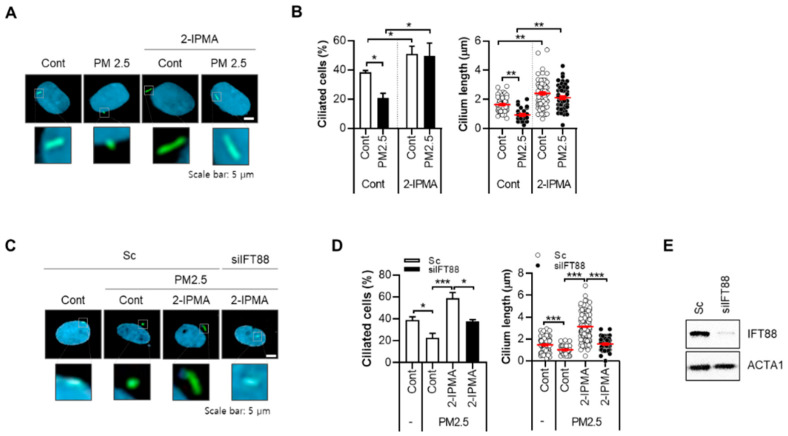
2-IPMA restores PM2.5-mediated ciliary dysgenesis in normal dermal fibroblasts. (**A**,**B**) Normal skin fibroblasts were treated with PM2.5 (50 µg/mL) for 24 h in the presence or absence of 2-IPMA (100 µM) for 24 h. (**A**) Then, the cells were imaged by immuno-staining with ARL13B antibody (green) and Hoechst dye (blue) Scale bar: 5 µm. (**B**) The ciliated cells and cilium length of the cells were measured under a fluorescence microscope. (**C**–**E**) Normal dermal fibroblasts transfected with either scrambled siRNA (Sc) or targeted siRNA to *IFT88* (si*IFT88*) were treated with PM2.5 (50 µg/mL) with or without 2-IPMA (100 µM) for 24 h. (**C**) Then, the cells were imaged by immuno-staining with ARL13B antibody (green) and Hoechst dye (blue), scale bar: 5 µm. (**D**) The ciliated cells and cilium length of the cells were measured. (**E**) The reduced expression of IFT88 by siRNA was analyzed by Western blotting. Data are presented as the mean ± SEM (*n* = 100, * *p* < 0.05, ** *p* < 0.01, *** *p* < 0.005).

**Figure 3 ijms-22-10941-f003:**
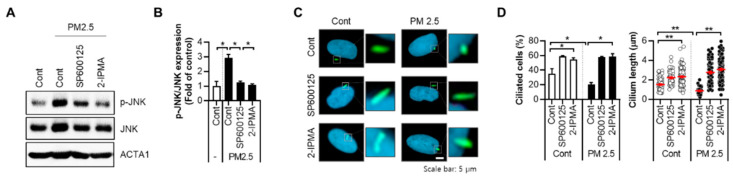
2-IPMA inhibits JNK activation by PM2.5 in normal dermal fibroblasts. (**A**,**B**) Normal dermal fibroblasts were treated with PM2.5 (50 µg/mL) with either 2-IPMA (100 µM) or SP600125 (10 µM) with 2-IPMA for 24 h. Then, the cells were harvested and analyzed by Western blotting with indicated antibodies (**A**). The relative expression of p-JNK verses JNK was analyzed using densitometry with the blots (**B**) (±SEM, *n* = 3, * *p* < 0.05). (**C**,**D**) Normal dermal fibroblasts were exposed to PM2.5 (50 µg/mL) with either 2-IPMA (100 µM) or SP600125 (10 µM) for 24 h. Then, the cells were stained with ARL13B antibody (green) and Hoechst dye (blue) (**C**). Both ciliated cells and cilium length of the cells were measured (**D**). Data are presented as the mean ± SEM (*n* = 100, * *p* < 0.05, ** *p* < 0.01). Scale bar: 10 µm.

**Figure 4 ijms-22-10941-f004:**
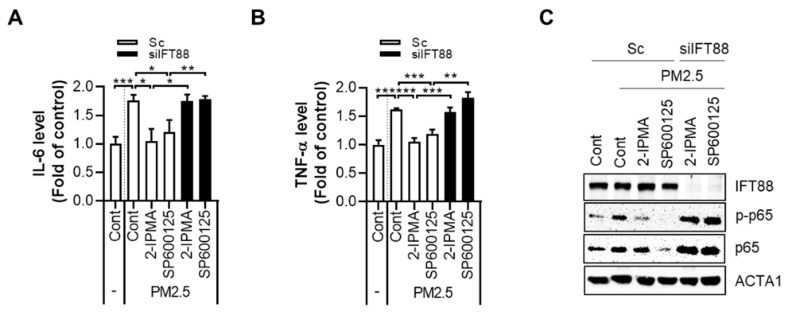
Primary cilia by 2-IPMA decrease inflammatory responses by PM2.5 in normal dermal fibroblasts. (**A**–**C**) Normal dermal fibroblasts transfected with either scrambled siRNA (Sc) or siRNA against *IFT88* (si*IFT88*) were further exposed to PM2.5 (50 µg/mL) with 2-IPMA (100 µM) or SP600125 (10 µM) for 24 h. Then, the expression levels of inflammatory cytokines, IL-6 (**A**) and TNF-α (**B**), in the culture condition were measured using an ELISA assay. The cells were harvested and analyzed by Western blotting with the indicated antibodies (**C**). Data are presented as the mean ± SEM (*n* = 3, * *p* < 0.05, ** *p* < 0.01, *** *p* < 0.005).

**Figure 5 ijms-22-10941-f005:**
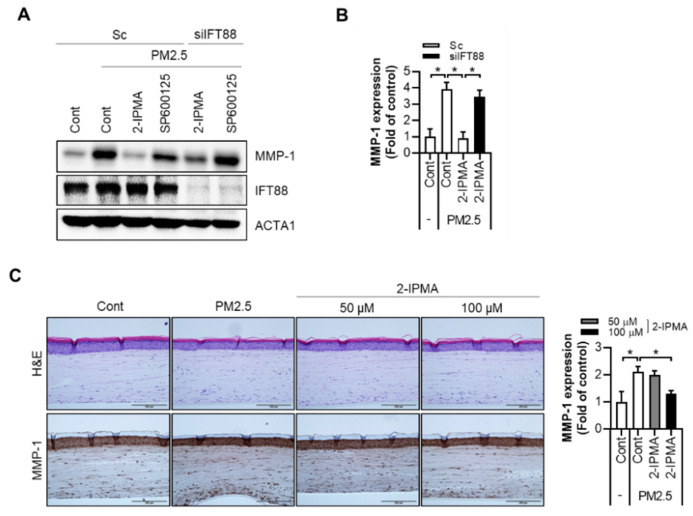
Primary cilia by 2-IPMA suppresses MMP-1 expression by PM2.5 in normal dermal fibroblasts and a 3-D-skin model. (**A**,**B**) Normal dermal fibroblasts transfected with either scrambled siRNA (Sc) or targeted siRNA to *IFT88* (si*IFT88*) were treated with PM2.5 (50 µg/mL) in the absence or presence of 2-IPMA (100 µM) or SP600125 (10 µM) for 24 h. Then, the cells were harvested and analyzed by Western blotting with the indicated antibodies (**A**). The expression of MMP-1 was quantified by densitometric image scanning of the Western blots (**B**) (*n* = 3, * *p* < 0.05). (**C**) Full-thickness human 3-D-skin models were treated with PM2.5 (100 µg/mL) in the absence or presence of 2-IPMA (50 or 100 µM) for 48 h. Then, the tissues were subjected to histological examination. (**C**) H&E staining (upper) and MMP-1 immunohistochemistry (lower). Quantification of MMP-1 expression in the dermis. Data are presented as the mean ± SEM (*n* = 3, * *p* < 0.05). Scale bar: 200 µm.

## Data Availability

Not applicable.

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
