# Peer review of "Primary Ciliogenesis by 2-Isopropylmalic Acid Prevents PM2.5-Induced Inflammatory Response and MMP-1 Activation in Human Dermal Fibroblasts and a 3-D-Skin Model"

_ijms, 2021, doi:10.3390/ijms222010941_

Round 1
Reviewer 1 Report
In the present work, the authors expand and continue their earlier research on the effects of fine particulate matters [PMs, (PM2.5 = diameters <2.5 μm)] on primary cilia development. They used 2-isopropylmalic acid (2-IPMA ) as inducer of primary ciliogenesis reversing negative effect of PM2.5 on primary cilia development. They also tried to explain potential mechanism underlying the observed effect and connected it with downregulation of inflammatory responses in normal dermal fibroblasts
The story is well written and seems to be interesting, the experimental model is quite comprehensive - including isogenic cells and 3D-skin model. I have some minor comments.
Photos of normal skin fibroblasts (Figure 1B and C for example) look similar form me. I cannot see difference presented on Figure 1E. The authors presented some results as Western Blot images. I would like to see originals. The authors stated “treatment with 2-IPMA additionally inhibits JNK phosphorylation in PM2.5-treated cell” (line 93), however no data and statistic have no provided. Similar, line 129 and Figure 4A. How did the authors normalize their data? Did the authors normalize to the specific “untreated control” within one single experiment, or was the data normalized to the average of the control values? The statistical section is superficial. Are assumptions of the ANOVA test met? Did Authors check normality and/or homogeneity of variance? If so what test did Authors use? If not all statistical analysis should be repeated. How post-hoc analysis was done?
Author Response
Response to Reviewers' comments: (Reviewer #1)
In the present work, the authors expand and continue their earlier research on the effects of fine particulate matters [PMs, (PM2.5 = diameters <2.5 μm)] on primary cilia development. They used 2-isopropylmalic acid (2-IPMA) as inducer of primary ciliogenesis reversing negative effect of PM2.5 on primary cilia development. They also tried to explain potential mechanism underlying the observed effect and connected it with downregulation of inflammatory responses in normal dermal fibroblasts. The story is well written and seems to be interesting, the experimental model is quite comprehensive - including isogenic cells and 3D-skin model. I have some minor comments.
Minor comments:
Q1. Photos of normal skin fibroblasts (Figure 1B and C for example) look similar form me. I can not see difference presented on Figure 1E.
Response 1: We appreciate your kind comment. Primary cilia are very tiny slender hair-like specialized organelle. Since the size is several microns long and 0.2 µm in diameter, it is hard to distinguish their size in cells. Thus, according to the reviewer’s kind comment, we added magnified image for each pictures (Figure 1B, D and 2B).
Q2. The authors presented some results as Western Blot images. I would like to see originals.
Response 2: Previously, all original images for the blots were submitted to Journal. However, according to the reviewer’s comments, we also added the images as Reviewer Figure 1. (Please check end of this text).
Q3. The authors stated “treatment with 2-IPMA additionally inhibits JNK phosphorylation in PM2.5-treated cell” (line 93), however no data and statistic have no provided. Similar, line 129 and Figure 4A.
Response 3: We appreciate your generous comment. According to the review’s suggestion, we added a statistic analysis for the Figure 2A and 4A.
Q4. How did the authors normalize their data? Did the authors normalize to the specific “untreated control” within one single experiment, or was the data normalized to the average of the control values? The statistical section is superficial. Are assumptions of the ANOVA test met? Did Authors check normality and/or homogeneity of variance? If so what test did Authors use? If not all statistical analysis should be repeated. How post-hoc analysis was done?
Response 4: We appreciate your kind valuable comment. For a statistic analysis, we normalized the data with ‘untreated control samples’. All statistical analysis was performed with ANOVA and student t-test and p values were described in each Figures and Material and Method part. We employed a Fisher’s least significant difference analysis for the post-hoc.
Reviewer Figure 1: The original images for the blots.

Reviewer 2 Report
The manuscript “Primary Ciliogenesis by 2-Isopropylmalic Acid Prevents PM2.5-Induced Inflammatory Response and MMP-1 Activation in Human Dermal Fibroblasts and 3D-Skin Model” presents the results of an experimental study performed to evaluate the possible beneficial effects of 2-isopropylmalic acid (2-IPMA), as an inducer for ciliogenesis, and to investigate influence of 2-IPMA on PM2.5-induced dysgenesis of primary cilia in dermal fibroblasts.
The data of this study show that 2-IPMA (an intermediate metabolite in leucine biosynthesis) inhibit the generation of excessive reactive oxygen species and the activation of stress kinase in PM2.5-treated dermal fibroblasts and inhibit the production of pro-inflammatory cytokines, including IL-6 and TNF-α, which were upregulated by PM2.5.
The subject is interesting and with relevance for the pathophysiological mechanisms and clinical impact and the results suggest that 2-IPMA can be a potential cosmetic agent of anti-pollution skincare.
The presentation of the aspects related to the study described in the paper is accompanied by corresponding bibliographical references. The obtained results are in accordance with the data from the specialized literature.
Recommend further research to understand the molecular mechanism of 2-IPMA in dermal fibroblasts.
Author Response
Response to Reviewers' comments: (Reviewer #2)
The manuscript “Primary Ciliogenesis by 2-Isopropylmalic Acid Prevents PM2.5-Induced Inflammatory Response and MMP-1 Activation in Human Dermal Fibroblasts and 3D-Skin Model” presents the results of an experimental study performed to evaluate the possible beneficial effects of 2-isopropylmalic acid (2-IPMA), as an inducer for ciliogenesis, and to investigate influence of 2-IPMA on PM2.5-induced dysgenesis of primary cilia in dermal fibroblasts. The data of this study show that 2-IPMA (an intermediate metabolite in leucine biosynthesis) inhibit the generation of excessive reactive oxygen species and the activation of stress kinase in PM2.5-treated dermal fibroblasts and inhibit the production of pro-inflammatory cytokines, including IL-6 and TNF-α, which were upregulated by PM2.5.
The subject is interesting and with relevance for the pathophysiological mechanisms and clinical impact and the results suggest that 2-IPMA can be a potential cosmetic agent of anti-pollution skincare. The presentation of the aspects related to the study described in the paper is accompanied by corresponding bibliographical references. The obtained results are in accordance with the data from the specialized literature. Recommend further research to understand the molecular mechanism of 2-IPMA in dermal fibroblasts.
Response: We very appreciate your kind comment and suggestion. We already obtained a patent 2-IPMA as a potential inducer for primary ciliogenesis and are planning to develop as a cosmetic ingredient for anti-pollution.
